# In Vivo Assessment of the Apatite-Forming Ability of New-Generation Hydraulic Calcium Silicate Cements Using a Rat Subcutaneous Implantation Model

**DOI:** 10.3390/jfb14040213

**Published:** 2023-04-11

**Authors:** Naoki Edanami, Shoji Takenaka, Razi Saifullah Ibn Belal, Kunihiko Yoshiba, Shintaro Takahara, Nagako Yoshiba, Naoto Ohkura, Yuichiro Noiri

**Affiliations:** 1Division of Cariology, Operative Dentistry and Endodontics, Department of Oral Health Science, Niigata University Graduate School of Medical and Dental Sciences, Niigata 951-8514, Japan; 2Division of Oral Science for Health Promotion, Department of Oral Health and Welfare, Niigata University Graduate School of Medical and Dental Sciences, Niigata 951-8514, Japan

**Keywords:** in vivo apatite-forming ability, rat subcutaneous implantation, new-generation hydraulic calcium silicate cements, micro-Raman spectrometry, electron probe micro-analyzer

## Abstract

Hydroxyapatite formation on endodontic hydraulic calcium silicate cements (HCSCs) plays a significant role in sealing the root canal system and elevating the hard-tissue inductivity of the materials. This study evaluated the in vivo apatite-forming ability of 13 new-generation HCSCs using an original HCSC (white ProRoot MTA: PR) as a positive control. The HCSCs were loaded into polytetrafluoroethylene tubes and implanted in the subcutaneous tissue of 4-week-old male Wistar rats. At 28 days after implantation, hydroxyapatite formation on the HCSC implants was assessed with micro-Raman spectroscopy, surface ultrastructural and elemental characterization, and elemental mapping of the material–tissue interface. Seven new-generation HCSCs and PR had a Raman band for hydroxyapatite (v1 PO_4_^3−^ band at 960 cm^−1^) and hydroxyapatite-like calcium-phosphorus-rich spherical precipitates on the surfaces. The other six HCSCs with neither the hydroxyapatite Raman band nor hydroxyapatite-like spherical precipitates did not show calcium-phosphorus-rich hydroxyapatite-layer-like regions in the elemental mapping. These results indicated that 6 of the 13 new-generation HCSCs possessed little or no ability to produce hydroxyapatite in vivo, unlike PR. The weak in vivo apatite-forming ability of the six HCSCs may have a negative impact on their clinical performance.

## 1. Introduction

Hydraulic calcium silicate cements (HCSCs) have been widely used in endodontics. The first commercial product of endodontic HCSC, grey ProRoot MTA (Dentsply, Tulsa, OK, USA), was launched in 1998. Subsequently, a tooth-colored variant, white ProRoot MTA (PR; Dentsply), was developed by reducing the iron oxide content of grey ProRoot MTA and was introduced to the market in 2002 [1]. A number of in vitro and in vivo investigations have shown that PR exhibits excellent biocompatibility, hard-tissue inductivity, and sealing capacity in the presence of moisture (such as blood and tissue fluids) [2,3,4]. Therefore, PR is currently regarded as the gold standard for endodontic treatments, including vital pulp therapy, root perforation repair, apexification, regenerative endodontics, and endodontic surgery [5].

PR induces hydroxyapatite formation on its surface when in contact with body fluids [6,7], and the favorable performance of PR in endodontic procedures is considered to be based on this apatite-forming ability [8,9]. The hydroxyapatite formed on PR contributes to the tight seal between the dentinal wall and the material by filling the interfacial gap [10]. Moreover, the hydroxyapatite supports the adhesion of hard tissue-forming cells to the material surface via preferential fibronectin adsorption [11], thereby promoting the formation of biological hard tissue barriers, which prevent bacterial invasion from the oral cavity to the dental pulp or periapical tissue [12,13,14].

Despite PR’s favorable biological and sealing properties, it has shortcomings, including a long setting time, handling difficulties, the ability to cause tooth discoloration, and the inclusion of heavy metals such as arsenic, chromium and lead [15,16]. In addition, the long setting time of PR is clinically inconvenient, which enhances the risk of material washout during placement [17]. Moreover, the potential for tooth discoloration attributable to the presence of bismuth oxide in PR restricts its use in the esthetic zone, such as the anterior maxillary teeth [18].

Recent advancements in endodontic technology have led to the development of new formulations of endodontic HCSCs (new-generation HCSCs) designed to overcome PR’s shortcomings [15]. Several modifications have been made to the new-generation HCSCs; for example, bismuth oxide has been replaced with other radiopacifiers (such as zirconium oxide, tantalum oxide, and calcium tungstate) to minimize the tooth discoloration potential. The Portland cement component containing heavy metal elements has been replaced by synthetic tri- and dicalcium silicate. In addition, calcium chloride and pozzolanic aluminosilicate compounds have been added to decrease the setting time, finer cement particles have been used to accelerate the setting, and thickening and plasticizer agents have been incorporated to improve the handling characteristics. Although PR requires mixing the powder and distilled water before use, some newer HCSCs are supplied with non-setting liquids that make the materials ready-to-use formulations.

The apatite-forming ability of new-generation HCSCs may differ from that of PR due to differences in their composition. To date, in vitro studies have shown that various HCSCs produce hydroxyapatite in artificial body fluids, such as Kokubo’s simulated body fluid [19], phosphate-buffered saline [20,21], and Hank’s balanced salt solution [22,23,24,25]. However, only a few studies have investigated the apatite-forming ability of new-generation HCSCs in vivo [6,26,27,28,29]. Furthermore, it has been shown that hydroxyapatite is more difficult to form on biomaterials under in vivo conditions compared with hydroxyapatite formation in artificial body fluids in vitro [29,30].

Therefore, this study aimed to evaluate the in vivo apatite-forming ability of 13 new-generation HCSCs, using PR as a positive control, in a rat subcutaneous implantation model. The null hypothesis of this study was that the in vivo apatite-forming ability of the tested new-generation HCSCs does not differ from that of PR.

## 2. Materials and Methods

### 2.1. Ethical Approval

All animal experiments were approved by the Committee on the Guidelines for Animal Experimentation of Niigata University (approval date: 28 April 2021; approval number: SA00912) and performed according to all applicable international, national, and institutional guidelines for the care and use of animals.

### 2.2. Materials

The new-generation HCSCs evaluated in this study included Endocem Zr (ECZr; Maruchi, Wonju, Korea), Biodentine (BD; Septodont, Saint-Maur-des-Fosses, France), MTA Flow (MFlow; Ultradent, South Jordan, UT, USA), MTA REPAIR HP (MTAHP; Angelus, Londrina, PR, Brazil), Endoseal MTA (EMTA; Maruchi), EndoSequence BC Sealer (EBC; Brasseler, Savannah, GA, USA), EndoSequence BC RRM Putty (EBCP; Brasseler), Well-Root ST (WST; Vericom, Chuncheon, Korea), Well-Root PT (WPT; Vericom), BioC Sealer (BioC; Angelus), BioC Repair (BioCR; Angelus), and Super MTA Paste (SMTA; Sun Medical, Moriyama, Japan). MFlow can be prepared in three different consistencies by changing the mixing ratios: a thin consistency for root resorption and apexification, a thick consistency for vital pulp therapy and perforation repair, and a putty consistency for the retrograde filling. We tested MFlow with a thin consistency (MFlow^thin^) and a putty consistency (MFlow^putty^). MFlow^thin^ was prepared by combining 0.19 g of powder with three drops of gel. MFlow^putty^ was prepared by mixing 0.19 g of powder with one drop of gel. PR was used as the positive control. The composition of the HCSCs is shown in Table 1.

### 2.3. Rat Subcutaneous Implantation

Four-week-old male Wistar rats (*n* = 21) weighing 70–80 g were purchased from Clea Japan (Tokyo, Japan). Of these, 14 rats were used for micro-Raman spectroscopy and surface ultrastructural and elemental characterization, whereas seven were used for elemental mapping. The rats were maintained under standardized conditions, including temperature (23 ± 2 °C), humidity (40–70%), 12 h light/dark cycle, ad libitum water access, and commercial pellet diet. Surgical procedures were performed under general anesthesia with medetomidine hydrochloride, midazolam, and butorphanol.

Four separate, 5-mm-long incisions were made on the back of each animal using No. 11 scalpel blades (Feather, Osaka, Japan), and surgical pockets were created by laterally extending the incisions. HCSCs were prepared according to the instructions and loaded into closed-end polytetrafluoroethylene tubes (length: 5 mm; outer diameter: 3 mm; inner diameter: 2 mm) (AS ONE, Osaka, Japan). The HCSC-filled tubes were immediately inserted into the surgical pockets, and the incisions were sutured using 4–0 silk (Mani, Tochigi, Japan). After surgery, the general health conditions of the animals were evaluated daily by veterinary technicians. No signs of distress were observed. At 28 days after surgery, the animals were euthanized with an isoflurane overdose, and the HCSC implants were collected with the surrounding tissue.

### 2.4. Micro-Raman Spectrometry

The HCSC implants with the adjacent connective tissue (*n* = 4 per material) were immersed in 6% sodium hypochlorite for 20 min to separate the connective tissue from the implants. After rinsing with distilled water, the HCSC-exposed surface of the implants was analyzed using a Raman spectrometer coupled to a microscope with a 100× objective lens (NRS-3100; JASCO, Tokyo, Japan). The Raman spectrometer was calibrated using a silicon standard at 520 cm^−1^. A laser beam with an excitation wavelength of 532 nm and a power of 7.4 mW was used. Measurements were performed at three random points for each sample; thus, 12 Raman spectra were taken per material. The Raman spectra of as-prepared HCSCs were also measured as negative controls.

### 2.5. Surface Ultrastructural and Elemental Characterization

The samples used for micro-Raman spectrometry were also used to analyze the surface ultrastructure and elemental composition of the in vivo-implanted HCSCs. The sample surfaces were coated with gold using an ion-sputtering device (IC-50; Shimadzu, Kyoto, Japan), and scanning electron microscope (SEM) images were obtained using an electron probe micro-analyzer (EPMA1601, Shimadzu) using an accelerating voltage of 15 kV. Elemental analyses were performed on spherical precipitates using the wavelength-dispersive X-ray spectroscopy (WDX) function of the EPMA1601. The beam spot size was 3 μm.

### 2.6. Elemental Mapping Analysis

Some HCSCs were further assessed regarding their hydroxyapatite formation after rat subcutaneous implantation using SEM-elemental mapping, as performed in our previous studies [6,29].

PR (positive control), ECZr, MTAHP, EMTA, WST, WPT, and BioCR implants (*n* = 4 for each material) with the surrounding tissue were fixed in 2.5% glutaraldehyde solution buffered with 60 mmol/L HEPES for 24 h at 4 °C. The fixed specimens were then dehydrated with increasing ethanol and acetone and embedded in methacrylate resin (Osteoresin; Wako, Osaka, Japan). The resin-embedded specimens were longitudinally sectioned through the center of the HCSC implants with a water-cooled diamond wheel saw (MC-201N; Maruto, Tokyo, Japan), polished with SiC paper to 4000-grit, and gold-coated with the IC-50 ion coater. Calcium (Ca)-, phosphorus (P)-, bismuth (Bi)-, tungsten (W)-, and zirconium (Zr)-element distributions of the material–tissue interface were mapped using the EPMA1601 at 1000× magnification. A representative area (226.8 μm × 226.8 μm) near the center of the tube opening was evaluated for each sample. The EPMA accelerating voltage was 15 kV, the step size was 1.8 μm, and the sampling time was 0.1 s at each point.

## 3. Results

Table 2 provides a summary of the results of this study.

### 3.1. Micro-Raman Spectrometry

Hydroxyapatite exhibits Raman spectra with a peak at 960 cm^−1^ (v1 PO_4_^3−^ band) [31]. Figure 1 shows representative Raman spectra recorded on the HCSC implants and as-prepared HCSCs. No Raman band was detected at 960 cm^−1^ on any of the as-prepared HCSCs. A 960 cm^−1^ Raman band attributable to hydroxyapatite was detected on the PR, BD, MFlow^thin^, MFlow^putty^, EBC, EBCP, BioC, and SMTA implants. However, the Raman spectra of the ECZr, MTAHP, EMTA, WST, WPT, and BioCR implants had no peak at 960 cm^−1^. A Raman band attributable to SO_3_^2−^ at 962 cm^−1^ [32,33] was detected on the as-prepared and in vivo-implanted ECZr, EMTA, and WPT. A Raman band attributable to calcite at 1086 cm^−1^ [34] was detected on all HCSC implants except the ECZr implants.

**Figure 1 jfb-14-00213-f001:**
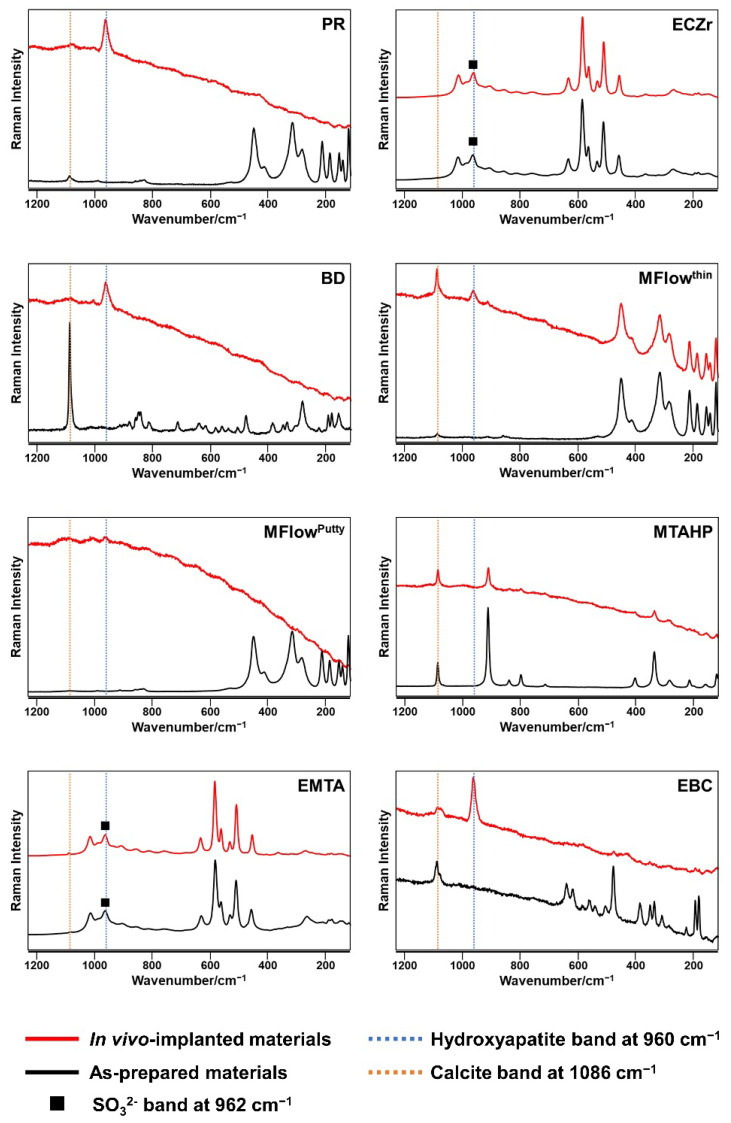
Representative Raman spectra of white ProRoot MTA (PR), Endocem Zr (ECZr), Biodentine (BD), MTA Flow with a thin consistency (MFlow^thin^), MTA Flow with a putty consistency (MFlow^putty^), MTA REPAIR HP (MTAHP), Endoseal MTA (EMTA), EndoSequence BC Sealer (EBC), EndoSequence BC RRM Putty (EBCP), Well-Root ST (WST), Well-Root PT (WPT), BioC Sealer (BioC), BioC Repair (BioCR), and Super MTA Paste (SMTA).

### 3.2. Surface Ultrastructural and Elemental Characterization

Hydroxyapatite-like spherical precipitates were detected with the SEM–WDX on the PR, BD, MFlow^thin^, MFlow^putty^, EBC, EBCP, BioC, and SMTA implants (Figure 2a) while not detected on the ECZr, MTAHP, EMTA, WST, WPT, and BioCR implants. Figure 2b shows the elemental composition detected by the spot analyses of the spherical precipitates. Ca, P, oxygen (O), and carbon (C) elements were detected on the spots, with a trace of other elements.

**Figure 2 jfb-14-00213-f002:**
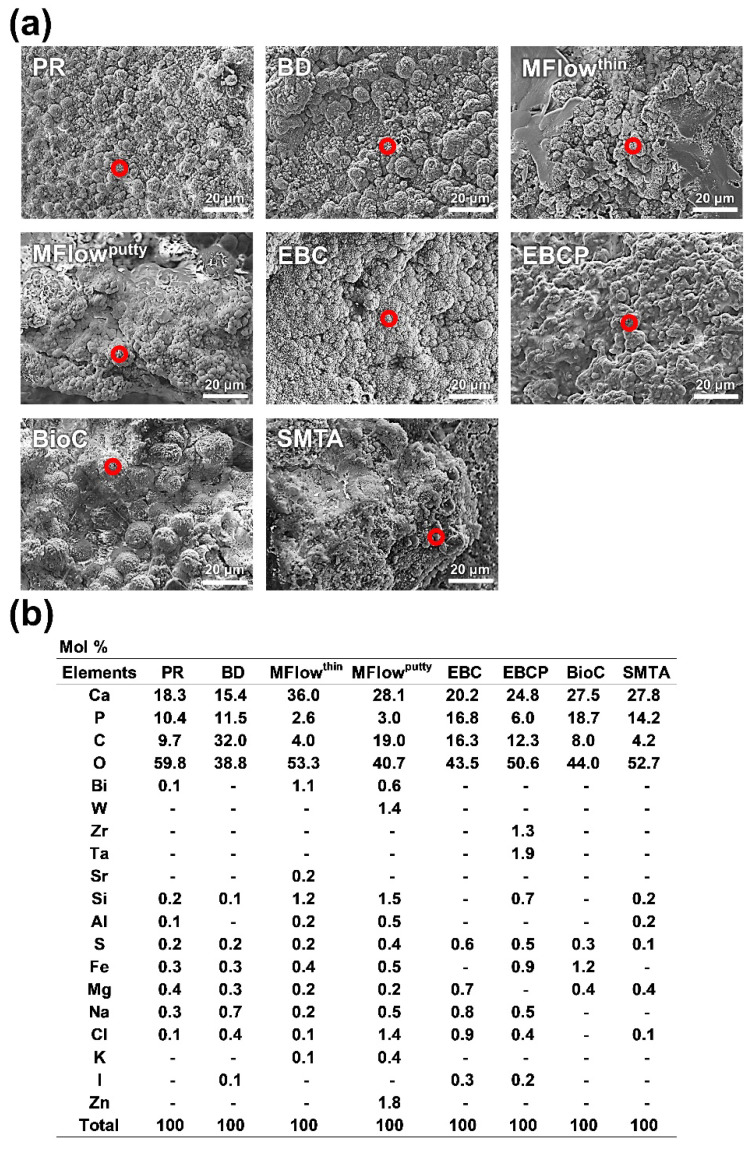
Scanning electron microscope images (**a**) and elemental compositions (**b**) of spherical precipitates formed on hydraulic calcium silicate cements after 28 days of implantation in rat subcutaneous tissue. Red circles indicate the areas for elemental analysis. All spherical precipitates contained calcium (Ca), phosphorus (P), carbon (C), and oxygen (O), which are constituents of carbonate-substituted hydroxyapatite. Scale marker = 20 μm. PR, White ProRoot MTA; BD, Biodentine; MFlow^thin^, MTA Flow with a thin consistency; MFlow^putty^, MTA Flow with a putty consistency; EBC, EndoSequence BC Sealer; EBCP, EndoSequence BC RRM Putty; BioC, BioC Sealer; SMTA, Super MTA Paste; Bi, bismuth; W, tungsten; Zr, zirconium; Ta, tantalum; Sr, strontium; Si, silicon; Al, aluminum; S, sulfur; Fe, iron; Mg, magnesium; Na, sodium; Cl, chlorine; K, potassium; I, iodine; Zn, zinc.

### 3.3. Elemental Mapping Analysis

Figure 3 shows representative SEM-elemental mapping images of the interfaces of the PR, ECZr, MTAHP, EMTA, WST, WPT, and BioCR implants with subcutaneous tissue. These HCSC implants contained radiopaque compounds indicated by the characteristic elements of Bi, W and Zr. Hydroxyapatite-like Ca-P-rich regions were detected on the PR implants. Such regions were not detected on the other HCSC implants.

**Figure 3 jfb-14-00213-f003:**
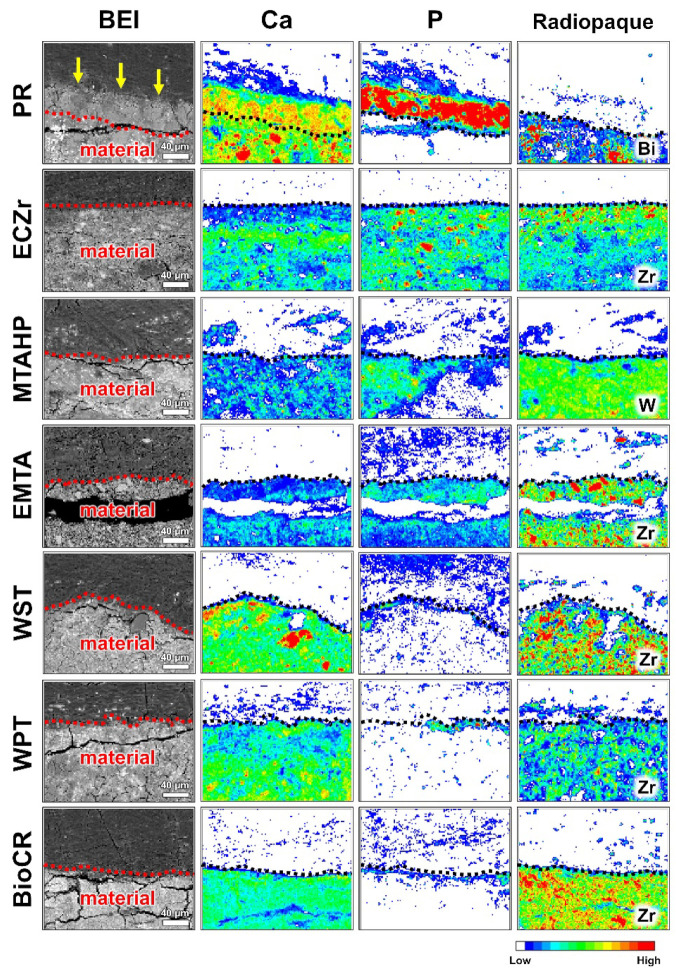
Elemental mapping assessment of the hydroxyapatite formation on white ProRoot MTA (PR), Endocem Zr (ECZr), MTA Repair HP (MTAHP), Endoseal MTA (EMTA), Well-Root ST (WST), Well-Root PT (WPT), and BioC Repair (BioCR) after 28 days of implantation in rat subcutaneous tissue. Representative backscattered electron images (BEI) and elemental mapping images of the material–tissue interface are shown. Note that the material is at the bottom, and the tissue is in the upper part of the images. A hydroxyapatite layer-like calcium (Ca)- and phosphorus (P)-rich region (indicated by arrows) is present on the PR implant. The hydroxyapatite layer-like region is absent on the ECZr, MTAHP, EMTA, WST, WPT, and BioCR implants. The lines are drawn between the radiopacifier (Bi, bismuth; W, tungsten; Zr, zirconium)-present and -absent regions. Color is the key to showing the intensity of the elements. Scale marker = 40 μm.

## 4. Discussion

PR, an original HCSC, forms hydroxyapatite in vivo [6,7]. A series of reactions occur after PR encounters body fluids. Initially, the hydration of tri- and dicalcium silicate in PR generates calcium silicate hydrate and calcium hydroxide. Subsequently, the high pH of calcium hydroxide promotes the dissolution of the surface of calcium silicate hydrate, leading to the formation of a negatively charged silica gel layer on the surface. The silica gel layer attracts calcium and phosphate ions from the surrounding environment, facilitating hydroxyapatite precipitation on the layer [35,36,37].

Hydroxyapatite formation on PR reduces leakage at the material–dentin interface. In a previous study, PR exhibited significantly higher sealing ability than the traditional root-end filling material, super EBA cement [38]. In addition, hydroxyapatite formation on PR facilitates the attachment of hard tissue-forming cells to the material surface. It is well documented that PR causes osteogenesis and cementogenesis directly on the material [13,14,39].

A number of new-generation HCSCs have been introduced as potential replacements for PR. These newer HCSCs have short setting times, minimal tooth discoloration potential, and/or improved handling properties compared to PR [15]. However, the modifications made to these HCSCs could compromise their apatite-forming ability.

We evaluated 13 new-generation HCSCs. ECZr, BD, MFlow^thin^, MFlow^putty^, and MTAHP are powder–liquid type HCSCs with short to medium setting times [40,41,42]. EMTA, EBC, EBCP, WST, WPT, BioC, and BioCR are ready-to-use HCSCs with a paste or putty consistency. SMTA is a resin-modified paste–catalyst type HCSC with improved operability, low solubility, and stable compressive strength [43,44].

These HCSCs and PR (positive control) were implanted in rat subcutaneous tissue for 28 days to evaluate their in vivo apatite-forming ability. The rat subcutaneous implantation model is an effective method for this purpose because the cells in subcutaneous tissue do not deposit biological hard tissues (such as dentin, bone, or cementum) that would interfere with the detection of hydroxyapatite generated by chemical reactions between body fluids and HCSCs.

Hydroxyapatite formation on the HCSC implants was qualitatively assessed by micro-Raman spectroscopy, surface characterization using SEM-WDX, and EPMA-elemental mapping. The PR, BD, MFlow^thin^, MFlow^putty^, EBC, EBCP, BioC, and SMTA implants exhibited a Raman band at 960 cm^−1^ attributable to hydroxyapatite (Figure 1) and showed hydroxyapatite-like Ca-P-rich spherulites on the surfaces (Figure 2). These findings indicate that these seven HCSCs produced hydroxyapatite in rat subcutaneous tissue. In contrast, the ECZr, MTAHP, EMTA, WST, WPT, and BioCR implants showed neither the Raman band for hydroxyapatite nor hydroxyapatite-like Ca-P-rich spherulites on their surfaces (Figure 1 and Figure 2). Moreover, the elemental mapping analysis demonstrated the absence of hydroxyapatite-like Ca-P-rich, radiopacifier-free regions on the ECZr, MTAHP, EMTA, WST, WPT, and BioCR implants (Figure 3). These results indicate that these six HCSCs formed no or very little hydroxyapatite in rat subcutaneous tissue, unlike PR. Therefore, the null hypothesis of this study that the in vivo apatite-forming ability of the tested new-generation HCSCs does not differ from that of PR was rejected.

Micro-Raman spectroscopy and SEM–WDX provided clear evidence of hydroxyapatite formation on the BD, MFlow^thin^, MFlow^putty^, EBC, EBCP, BioC, and SMTA implants. Thus, we did not further evaluate these implants using elemental mapping. However, it can be hypothesized that elemental mapping of these implants would have revealed a Ca-P-rich layer composed of hydroxyapatite.

Our method might lack the sensitivity required to detect extremely thin layers of hydroxyapatite (less than a few microns). According to the standard ISO 23317 (in vitro evaluation for the apatite-forming ability of implant materials) [45], thin-film XRD is recommended for detecting such thin hydroxyapatite layers formed on biomaterials. In this regard, thin-film XRD could yield more sensitive results concerning hydroxyapatite detection on the HCSC implants. However, micro-Raman spectroscopy is a powerful tool for detecting hydroxyapatite. A previous study reported that this technique detected 0.1 wt% hydroxyapatite in soils [46]. Moreover, the elemental mapping of the current research at ×1000 magnification should be able to detect Ca–P-rich layers thicker than approximately 5 µm. Thus, our findings are sufficient to conclude that ECZr, MTAHP, EMTA, WST, WPT, and BioCR formed no or very little hydroxyapatite on their surfaces.

In this study, the PR implants had a thick hydroxyapatite layer (approximately 60 µm thick) on the surfaces (Figure 3). In a previous study, hydroxyapatite-like precipitates appeared on PR within 12 h in rat subcutaneous tissue [7]. Therefore, the current findings that ECZr, MTAHP, EMTA, WST, WPT, and BioCR implants did not show hydroxyapatite formation after 28 days in rat subcutaneous tissue indicate these six HCSCs had considerably weaker in vivo apatite-forming ability than PR.

Previous studies have assessed the biomineralization ability of HCSCs in vivo using the von Kossa birefringence technique [47,48,49]. However, this method only proves if calcite (CaCO_3_) formed, indicating the calcium hydroxide reaction product became carbonated in vivo. Other techniques, such as elemental analysis or spectroscopic analysis, are necessary for detecting hydroxyapatite-like calcium phosphate precipitates. This study used micro-Raman spectroscopy, SEM–WDX, and EPMA-elemental mapping to detect hydroxyapatite.

The in vivo apatite-forming ability of EBC and EBCP has been investigated. A study using an electron probe micro-analysis reported that EBC formed hydroxyapatite-like precipitates in rat subcutaneous tissue [29]. Another study detected hydroxyapatite-like precipitates using SEM-energy dispersive X-ray spectrometry on in vivo-implanted EBC [28]. These previous findings are consistent with our finding that EBC formed hydroxyapatite in vivo. Moinzadeh et al. reported a surgical endodontic case in which EBCP used for retrograde-filling did not show hydroxyapatite formation [26]. This finding conflicts with our finding that EBCP displayed hydroxyapatite formation in vivo. However, this clinical case is a failure case. Persistent inflammation in the failure case may have created an acidic environment unfavorable for hydroxyapatite nucleation [50,51]. Therefore, in successful clinical cases without persistent inflammation, EBCP may produce hydroxyapatite, in line with the current study’s findings.

Previous studies have reported that in vitro experiments using artificial body fluids do not accurately predict the ability of HCSCs to produce hydroxyapatite in vivo [26,27,29]. Some discrepancies were found between the previous in vitro and current in vivo results regarding hydroxyapatite formation. For example, ECZr, MTAHP, and WST, which had produced hydroxyapatite-like precipitates in artificial body fluids within a short time in vitro [19,20,22], did not show hydroxyapatite formation after 28-day implantation in rat subcutaneous tissue. The discrepancies may be caused by the absence of proteins in the artificial body fluids that are present in real body fluids. Proteins, especially albumin, disrupt hydroxyapatite formation on biomaterials by blocking the apatite nucleation sites on the materials (such as silanol groups) [52,53,54].

The therapeutic effects of HCSCs may depend on the level of apatite-forming ability because the hydroxyapatite formed on HCSCs enhances the material’s sealing ability and hard-tissue inductivity [10,11]. In this study, ECZr and WPT did not form a detectable amount of hydroxyapatite in vivo, while PR did. The differences in apatite-forming ability may explain the less favorable outcomes of ECZr and WPT compared with the outcomes using PR in vital pulp therapy [55,56]. Additionally, we showed that MTAHP, EMTA, WST, and BioCR also did not form a detectable amount of hydroxyapatite in rat subcutaneous tissue. These materials could be less effective in promoting the healing of dental pulp and periapical tissue than PR.

This study found that seven new-generation HCSCs have the ability to form hydroxyapatite in vivo, whereas six new-generation HCSCs have little or no such ability. These findings aid in designing new HCSCs that exhibit favorable in vivo apatite-forming ability.

Two previous studies found that incorporating tricalcium aluminate in tricalcium silicate cement at concentrations >10% reduces its ability to form hydroxyapatite in artificial body fluid in vitro [57,58]. In this study, all aluminum-free HCSCs (BD, EBC, and EBCP) formed hydroxyapatite in vivo. Moreover, tricalcium aluminate-rich MTAHP containing 5–12% tricalcium aluminate failed to form a detectable amount of hydroxyapatite in vivo despite its high content of tricalcium silicate (45–55%) and dicalcium silicate (10–15%), both of which encourage hydroxyapatite formation. Therefore, reducing the content of tricalcium aluminate in HCSCs might enhance their potential to form hydroxyapatite in vivo.

In this study, BioC, but not BioCR, displayed hydroxyapatite formation, despite the identical composition of these materials, with the sole difference being the higher concentration of polyethylene glycol in BioC [59]. A previous study reported that BioC released more calcium and silicate ions than BioCR [59]. The enhanced ion release capacity of BioC may be the reason for the observed hydroxyapatite formation following subcutaneous implantation in rats.

Calcite was detected on all tested HCSCs except ECZr after in vivo implantation. Calcite has been found to bind to fibronectin, like hydroxyapatite [60]. However, calcite is more susceptible to dissolution in biological environments than hydroxyapatite [61]. The potential role of calcite in sealing the root canal system or in healing the pulp and periapical tissue warrants further examination.

A limitation of this study was the use of subcutaneous tissue of rats. In clinical use, the HCSCs contact dental pulp or periapical tissue. Therefore, future studies should evaluate the in vivo apatite-forming ability of HCSCs in clinically relevant tissues such as pulp tissue [62,63] or periapical tissue [13,64,65].

## 5. Conclusions

PR, BD, MFlow^thin^, MFlow^putty^, EBC, EBCP, BioC, and SMTA had the ability to produce hydroxyapatite after implantation in rat subcutaneous tissue. Conversely, ECZr, MTAHP, EMTA, WST, WPT, and BioCR had little or no ability to produce hydroxyapatite after implantation in rat subcutaneous tissue. The weak in vivo apatite-forming ability of ECZr, MTAHP, EMTA, WST, WPT, and BioCR may have a negative impact on their clinical performance.

## Figures and Tables

**Table 1 jfb-14-00213-t001:** Materials used in this study.

Material (Abbreviation)	Composition (wt%)
White ProRoot MTA (PR)	Powder: Portland cement (tricalcium silicate, dicalcium silicate, tricalcium aluminate, and other trace compounds) (60–90%), bismuth oxide (10–40%), and gypsum
	Liquid: distilled water
Endocem Zr (ECZr)	Powder: natural pure cement (calcium oxide, silicon dioxide, aluminum oxide, and other mineral oxides) (50%) and zirconium oxide (50%)
	Liquid: distilled water
Biodentine (BD)	Powder: tricalcium silicate, dicalcium silicate, calcium carbonate (10–25%), zirconium oxide, calcium oxide, and iron oxide
	Liquid: calcium chloride, hydrosoluble polymer, and distilled water
MTA Flow (MFlow)	Powder: tricalcium silicate (<50%), bismuth trioxide (<30%), dicalcium silicate (<20%), calcium sulfate (<3%), and silica (<0.1%)
	Liquid: water-soluble silicone-based gel
MTA REPAIR HP (MTAHP)	Powder: tricalcium silicate (45–55%), calcium tungstate (20–30%), dicalcium silicate (10–15%), tricalcium aluminate (5–12%), and calcium oxide (1–5%)
	Liquid: distilled water and organic plasticizer
Endoseal MTA (EMTA)	Ready-to-use paste: radiopacifiers (zirconium oxide and bismuth oxide) (47.28%), natural pure cement (calcium oxide, silicon dioxide, aluminum oxide, and other mineral oxides) (27.81%), and thickening agents (24.91%)
EndoSequence BC Sealer (EBC)	Ready-to-use paste: zirconium oxide (35–45%), tricalcium silicate (20–35%), dicalcium silicate (7–15%), calcium hydroxide (1–4%), calcium phosphate monobasic, filler, and thickening agents
EndoSequence BC RRM Putty (EBCP)	Ready-to-use putty: tricalcium silicate (30–36%), zirconium oxide (15–18%), tantalum pentoxide (12–15%), dicalcium silicate (9–13%), calcium sulfate (3–8%), calcium phosphate monobasic, filler, and thickening agents
Well-Root ST (WST)	Ready-to-use paste: calcium aluminosilicate compound, zirconium oxide, filler, and thickening agents
Well-Root PT (WPT)	Ready-to-use putty: calcium aluminosilicate compound, zirconium oxide, filler, and thickening agents
BioC Sealer (BioC)	Ready-to-use paste: tricalcium silicate, dicalcium silicate, tricalcium aluminate, calcium oxide, zirconium oxide, silicon oxide, polyethylene glycol, and iron oxide
BioC Repair (BioCR)	Ready-to-use putty: tricalcium silicate, dicalcium silicate, tricalcium aluminate, calcium oxide, zirconium oxide, silicon oxide, polyethylene glycol, and iron oxide
Super MTA Paste (SMTA)	Paste: Portland cement (tricalcium silicate, dicalcium silicate, tricalcium aluminate, and other trace compounds) (30–40%), zirconium oxide (30–40%), and hydroxypropyl methacrylate (20–30%)
	Catalyst: partially oxidized tri-n-butylborane, n-hexane, and ethanol

The compositions of the agents were in accordance with the information disclosed by the manufacturers. The thickener components of EMTA, EBC, EBCP, WST, and WPT were not disclosed.

**Table 2 jfb-14-00213-t002:** Summary of the study results.

Material	Abbreviation	HA Detected via Micro-Raman? (Figure 1)	Calcite Detected via Micro-Raman? (Figure 1)	HA-Like Spherulites Detected via SEM-WDX? (Figure 2)	Ca-P-Rich Layer Detected in Elemental Mapping? (Figure 3)
White ProRoot MTA	PR	**yes**	**yes**	**yes**	**yes**
Endocem Zr	ECZr	no	no	no	no
Biodentine	BD	**yes**	**yes**	**yes**	-
MTA Flow with a thin consistency	MFlow^thin^	**yes**	**yes**	**yes**	-
MTA Flow with a putty consistency	MFlow^putty^	**yes**	**yes**	**yes**	-
MTA REPAIR HP	MTAHP	no	**yes**	no	no
Endoseal MTA	EMTA	no	**yes**	no	no
EndoSequence BC Sealer	EBC	**yes**	**yes**	**yes**	-
EndoSequence BC RRM Putty	EBCP	**yes**	**yes**	**yes**	-
Well-Root ST	WST	no	**yes**	no	no
Well-Root PT	WPT	no	**yes**	no	no
BioC Sealer	BioC	**yes**	**yes**	**yes**	-
BioC Repair	BioCR	no	**yes**	no	no
Super MTA Paste	SMTA	**yes**	**yes**	**yes**	-

Hyphens indicate “not assessed”. HA, hydroxyapatite; SEM–WDX, scanning electron microscopy–wavelength-dispersive X-ray spectroscopy; Ca, calcium; P, phosphorus.

## Data Availability

The datasets generated and/or analyzed during the current study are available from the corresponding author upon reasonable request.

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
