# Peer review of "In Vivo Assessment of the Apatite-Forming Ability of New-Generation Hydraulic Calcium Silicate Cements Using a Rat Subcutaneous Implantation Model"

_jfb, 2023, doi:10.3390/jfb14040213_

Round 1

Reviewer 1 Report

Title paper “In Vivo Assessment of the Apatite-Forming Ability of Second-Generation Hydraulic Calcium Silicate Cements Using a Rat Subcutaneous Implantation Model”

The paper appears interesting and appropriate for the journal. It analyses in vivo the apatite forming ability of several second-generation calcium silicate based or containing endodontic sealers. The methodology is adequate but the back ground and the discussion must be deeply modified.

In the introduction and discussion, the Authors must cite and discuss the FIRST milestone papers published on CaSi materials, the first papers that innovatively introduced the methodologies to test the apatite-forming ability and the osteoinductive properties of CaSi cements.

Specifically, the Authors must cite the most productive and outstanding groups on Calcium Silicate cements using Web of Science (WOS) at the https://www.webofscience.com/wos/woscc/basic-search using the keywords:

# Calcium silicate cements + endodontics 
# apatite forming ability + calcium silicate cements
# bioactivity + endodontics

# Calcium silicate cements + Micro-Raman

# apatite forming ability + ESEM-EDX

The papers that report the drawbacks of Pro-Root MTA, including the setting time and the tooth-discoloration induced by Bismuth oxide must be cited and discussed.

In the discussion, the osteoinductive properties of CaSi based materials should be discussed taking into account some animal model studies:

Gandolfi, M. G., Iezzi, G., Piattelli, A., Prati, C., & Scarano, A. (2017). Osteoinductive potential and bone-bonding ability of ProRoot MTA, MTA Plus and Biodentine in rabbit intramedullary model: Microchemical characterization and histological analysis. Dental materials: official publication of the Academy of Dental Materials33(5), e221–e238. https://doi.org/10.1016/j.dental.2017.01.017

The reference 24 (Niu et al 2014) reported the reactions of CaSi hydration. Authors should replace it and cite better the reactions, that are more deeply described in the following papers:

Taddei P, Tinti A, Gandolfi MG, Rossi PL, Prati C. Vibrational study on the bioactivity of Portland cement-based materials for endodontic use. J Mol Struct 2009;924-926(C):548-554.

We ask to review the revised version of the paper after author resubmission.

Reviewer 2 Report

Dear authors,

The study is very interesting in the sense of the topic as well as the working method approached.

I don't have many recommendations, except those regarding the abstract, which must be more concisely written and the conclusions must correspond to those in the article.

The introduction is well written and the information is concise, to the point.

Results: In figures 2 and 3, I think it would be good to have a measuring scale.

Discussions and Conclusions are well written.

Reviewer 3 Report

This is an interesting study but some of it is confusing, which reduces its impact.

Materials that are used surgically and those for orthograde root canal treatment were combined, and all were denoted as 2nd generation materials The authors need to think about what "2nd generation" means. 2nd generation is not universally defined, but could mean cement materials without bismuth oxide.  Others view premixed  paste materials as second generation. And certainly, the materials  used for endodontic sealers with gutta percha, are yet another set of materials that can be considered a generation- or some other distinguishing term.  

Table 1 should have listed the components on the SDS sheets for the materials because calcium silicate and calcium aluminate are non-specific. Many calcium silicate and calcium aluminate CEMENT compounds exist: tricalcium silicate, dicalcium silicate, tricalcium aluminate calcium monoaluminate calcium dialuminate.....  The components should be listed from greatest to least- including the phases. These details are important for this study. 

Table 1 should identify all the materials  that were presented as a premixed paste. Please note that for the pastes, the liquid components were unidentified in Table 1 (except PEG in the BioC), as were the "thickeners", but these unidentified components represent significant constituents, that may be pertinent to this study.

The authors don't demonstrate comprehensive understanding of the materials and their differences. Their Table 1 overlooks important comparisons that they could have gleaned from the manufacturer's data.

Also, MTA Flow is not indicated for endodontic sealing, so the mixture should not have been mixed in a low viscosity consistency. 

MTA Fillapex should not be included- it is a resin-based material with a minor content of calcium silicate cement- and certainly different than the other "so-called" 2nd generation materials.  

The use of the abbreviations that are so similar was very confusing versus product names. A comprehensive table is needed, and a sample table  is attached that the reviewer used to understand the text.

The mention of the pozzolanic effect is offhand, and not fully discussed for its importance for HA formation, or to which material this would apply.

No mention was made of comparing the paste or putty products versus the powder/liquid products. 

Why would the two BioC materials differ? They are " identical" in Table 1.

What were the "other" elements? Sulfur may be important.   Why is Ta present in Biodentine? Why was it absent in the EBC or EBCP?

Why were only 8 interfaces mapped- this should be explicit in the text.

According to ISO 23317, a thin layer of HA may be formed and the document  suggests thin film XRD for detection. Perhaps the crossections were at too low a magnification to see layers of 1 to 3 µm thick.  

The conclusion that these materials are weaker in HA formation and therefore ineffective clinically is too strong for these results.  

The authors need to rewrite the manuscript with these comments considered and incorporated.  See also the attached files.

Round 2

Reviewer 1 Report

The paper can now be accepted.

Reviewer 3 Report

Some minor changes are needed; See attached.
